# Antinociceptive, Sedative and Excitatory Effects of Intravenous Butorphanol Administered Alone or in Combination with Detomidine in Calves: A Prospective, Randomized, Blinded Cross-Over Study

**DOI:** 10.3390/ani13121943

**Published:** 2023-06-09

**Authors:** Ekaterina Gámez Maidanskaia, Alessandro Mirra, Emma Marchionatti, Olivier Louis Levionnois, Claudia Spadavecchia

**Affiliations:** 1Anesthesiology and Pain Therapy Division, Department of Clinical Veterinary Medicine, Vetsuisse Faculty, University of Bern, 3012 Bern, Switzerland; alessandro.mirra@unibe.ch (A.M.); olivier.levionnois@unibe.ch (O.L.L.); claudia.spadavecchia@unibe.ch (C.S.); 2Clinic for Ruminants, Department of Clinical Veterinary Medicine, Vetsuisse Faculty, University of Bern, 3012 Bern, Switzerland; emma.marchionatti@unibe.ch

**Keywords:** calves, butorphanol, detomidine, antinociception, trigeminocervical reflex, excitation, sedation

## Abstract

**Simple Summary:**

Surgical procedures are routinely performed in calves, both under sedation and general anesthesia. The availability of analgesic drugs is limited in food-producing animals, making their perioperative management challenging. Few are the studies assessing the extent of pharmacological antinociception obtained when administering sedative and analgesics in cattle and the tools used are mainly qualitative. With the present study, we aimed at assessing the antinociceptive effect of butorphanol with or without detomidine in calves, using the modulation of the trigeminocervical reflex threshold as the main outcome. The trigeminocervical reflex threshold is a quantifiable nociceptive reflex that can be used to assess the modulation of pain processing. As secondary aims, physiological and behavioral effects of the administered treatments were evaluated, and the temporal profile of drug activity was described. This study provides quantitative evidence of the efficacy of butorphanol administered alone or combined with detomidine in calves. The presented findings can now contribute to optimizing the application of these drugs in clinical practice.

**Abstract:**

(1) Background: The diagnostic and therapeutic procedures performed under sedation or general anesthesia in bovines are numerous. The analgesic drugs that can be legally used are few, making perioperative analgesia challenging. (2) Methods: Calves were administered butorphanol 0.1 mg kg^−1^ alone (SB) or combined with 0.02 mg kg^−1^ of a detomidine (DB) IV. The antinociceptive effect (trigeminocervical reflex threshold (TCRt)), as well as the behavioral (sedation and excitation) and physiological (heart and respiratory rate) changes were investigated. Five time windows were defined: BL (30 min pre-injection), T1 (0–30 min post-injection (PI)), T2 (31–60 min PI), T3 (61–90 min PI) and T4 (91–120 min PI). (3) Results: Both groups had a significative increase in TCRt at T1-T4 compared to the BL. The TCRt was significatively higher in DB than in SB at T1, T2 and T4. Heart rate decreased significatively in DB compared to that in BL. Calves were significantly more sedated in the DB group, and significantly more excited in the SB group compared to the BL. (4) Conclusions: Butorphanol alone has a statistically significant antinociceptive effect, but it elicits marked excitation, limiting its clinical applicability under this dosing regimen. The co-administration of detomidine eliminated the excitatory effect and induced consistent sedation and a significantly more pronounced antinociceptive effect.

## 1. Introduction

Several diagnostic and therapeutic procedures are performed in bovines under sedation or general anesthesia [1,2,3,4,5]. Since only a few analgesic drugs can be legally used in food-producing animals (Commission Regulation (EU) no. 37/2010), their perioperative pain management is particularly challenging.

The use of alpha-2 agonists alone or combined with opioids have been described for several procedures, as castration or disbudding [6,7], in order to reduce stress associated with physical restraint or for anesthetic requirements [4,5,8]. Detomidine is a potent alpha-2 agonist registered for use in cattle. Its sedative and analgesic properties have been described for the drug used alone [9,10] or as a part of restrain protocols [3,4,11].

Butorphanol is a mixed agonist–antagonist opioid with both analgesic and sedative properties [12]. Few studies have investigated its analgesic activity in calves [5,13]. Results have shown that adding butorphanol to xylazine or/and lidocaine provides an increase in the duration of postoperative analgesia [5,12]. However, its sedative effects have been reported only when used in combination with alpha-2 agonists [4]. No study was found in the literature describing the anti-nociceptive, sedative and excitatory effects of butorphanol administered alone in calves, while, regarding other animal species, excitation was reported in piglets [14].

Analgesia elicited by alpha-2 agonist-based protocols in cattle has classically been assessed by determining the drug-induced modification of behavioral responses to nociceptive stimuli [7,9,15].

As an alternative, nociceptive reflexes, which can be precisely quantified with a neurophysiological approach, could be used to describe the efficacy profile of these drugs in bovines, as previously shown in other species [16]. Nociceptive reflexes are polysynaptic spinal reflexes that can be elicited by the electrical stimulation of relevant afferents and recorded via electromyography (EMG) from muscles involved in the nocifensive response to stimulation [17]. They can be used to perform a non-invasive, quantitative evaluation of antinociception evoked by analgesic drugs, providing specific information on their mechanisms of action and effect duration [18,19]. The trigeminocervical reflex (TCR), based on the recording of cervical muscle activity in response to trigeminal noxious stimulation, has been previously characterized in horses [20,21] and calves [22].

The primary aim of this study was to evaluate the antinociceptive activity provided by an intravenous bolus of butorphanol administered with or without detomidine, using the modulation of the trigeminocervical reflex threshold (TCRt) as the main outcome. 

As secondary aims, physiological (changes in heart rate and respiratory rate) and behavioral (sedation and/or excitation) effects of the administered treatments were evaluated, and their temporal profile was described. 

The primary hypothesis was that the TCRt would increase following butorphanol administration when compared to the baseline and that this increase would be more remarkable when butorphanol and detomidine were combined. The secondary hypothesis was that butorphanol would induce excitation, and that the co-administration with detomidine would allow it to be counteracted.

## 2. Materials and Methods

### 2.1. Animals

Five female and three male healthy calves were included in the study. At the beginning of the experiment, the median age was 59 (ranging from 24 to 69) days old and the median weight was 67 (ranging from 58–120) kg. Three of the calves were Limousins, two were Jerseys, two were Fleckviehs and one was a cross-breed. Health was assessed based on the clinical examination, as well as on hematology and blood chemistry results. The experiments were performed in the experimental facility of the Vetsuisse Faculty, University of Bern. The animal study protocol was approved by the Committee for Animal Experiments of the Canton of Bern Switzerland (permission number: 28719).

### 2.2. Study Design and Set-Up

The study was designed as a prospective, randomized, blinded, cross-over study. Two calves per day participated in the trial. The day before the experiment, they were transported from the farm of origin to the experimental facility, where they were housed together in a box with straw bedding (330 × 300 cm). Neither food nor water were withheld during their stay. The sites for electrode and jugular catheter placement were clipped on the day of arrival. The next morning, the two calves were brought into the experimental room. The first calf was placed in a box (63 × 163 × 100 cm), where it was kept during the entire experimental session. The second calf was placed in an adjacent area with straw bedding. Auditory and visual contact between the two animals was continuously allowed. Two intravenous catheters (14 G) were placed in the left and right jugular veins for treatment administration and blood sampling (intended for another study), respectively. Electrode sites were shaved and defatted. Self-adhesive surface electrodes (BlueSensor N, Ambu, Ballerup, Denmark) were used for stimulation and EMG recording. The cathode stimulation electrode was placed over the right supraorbital foramen, the anode was positioned approximately 1 cm rostrally to it (Figure 1A). A latex-free self-adhesive foam dressing (Animal Polster, Snøgg^®^, Solna, Sweden) was then placed over stimulating electrodes and cables to keep them on-site (Figure 1B). The EMG electrodes were placed on the contralateral cleidoccipitalis muscle (midway between the cranial aspect of the shoulder and the end of the maxilla, and approximately at the level of the transverse apophysis of the fourth cervical vertebra). The ground electrode was positioned approximately 50 mm dorsally to the EMG electrodes (Figure 1C). Self-adhesive foam and a cross-elastic cohesive bandage (Vetrap 3M^®^, Decatur, GA, USA) were placed around the neck (Figure 1D). Electrode impedance was checked automatically before and during each experimental session. In case of high impedance, electrodes were replaced. At the end of the experiment, the same procedures were performed on the second calf, while the first was left resting on the straw bedding. Both calves were brought back to the stall at the end of the second experiment. Measurements, scoring and treatment administration were performed by a single blinded investigator (EGM).

### 2.3. Sample Size Calculation

As no species-specific data were available on the extent of antinociceptive effects of opioids in calves, a sample size equal to the one used in Studer et al. was used [23]. In that study, the effects of levomethadone on the nociceptive withdrawal reflex (NWR) threshold were evaluated in 8 horses.

### 2.4. Treatments

Each calf received two treatments with a wash-out period of 2 weeks. The order of treatment administration was randomized (www.randomization.com, accessed on 15 December 2019). Calves received a 0.1 mg kg^−1^ butorphanol (Morphasol, 10 mg mL^−1^, Graeub, Bern, Switzerland) IV injected over 1 min combined with either a 0.02 mg kg^−1^ detomidine (Domosedan, 10 mg mL^−1^, Vetoquinol GmbH, Ismaning, Germany) IV (DB group) or an equal volume of physiologic saline solution (SB group). 

### 2.5. Data Collection

#### 2.5.1. TCRt Measurement

Electrical stimulation and electromyographic recordings to evaluate the TCRt were performed using a commercial unit (PainTracker; Dolosys GmbH, Berlin, Germany). Stimulation consisted of a train of five, 1 ms constant-current square-wave pulses delivered at a frequency of 200 Hz. The interstimulus interval was 10 s with 30% randomization and steps of maximum of 1mA were used for threshold tracking, starting at 1 mA. Electromyographic activity was recorded from 100 ms prior to until 400 ms after the stimulus for a total recording time of 500 ms (sampling frequency 1 kHz). The time window of interest for quantifying the TCRt was between 50 and 120 ms following stimulus onset. For noise quantification, a window between 130 and 10 ms prior to stimulation onset was assessed. The reflex was considered positive at an interval peak Z-score of above 10, meaning that the difference between the maximum EMG amplitude within the TCRt range and the mean EMG amplitude within the noise range had to be above 10 times the standard deviation of the EMG amplitudes within the noise range [24]. The TCRt was automatically and continuously determined through a validated threshold tracking algorithm [25]. The TCRt recording started at least 30 min before treatment administration and continued until the threshold returned to the baseline or restraint was no longer tolerated, until a maximum of 160 min after treatment administration. Off-line, an average TCRt was calculated for each minute of recording and the calculated values were then used for further statistical analysis. In order to simplify graphical representation, TCRt values were averaged over 5 min and are reported in Figure 2.

#### 2.5.2. Heart Rate and Respiratory Rate

Heart rate (HR) and an electrocardiogram (II lead ECG) were continuously recorded using a telemetric device (Televet 100, Kruuse, Langeskov, Denmark). Respiratory rate (RR) was determined by counting thoracic excursions per minute. Heart rate and respiratory rate were recorded on a purpose-made sheet every 5 min, during the whole experimental session.

#### 2.5.3. Sedation and Excitation

For the evaluation of sedation, an adapted version of the scale previously described by Ede et al. in 2019 [26] was used. The rating was 0 (no effect), 1 (lowered head, braced stance, or hindquarter weakness), 2 (sternal or lateral recumbency, some responsiveness to environment) or 3 (sternal or lateral recumbency, no responsiveness to environment). For the evaluation of excitation, a scale based on behavioral items was created on purpose (the behaviors evaluated were pushing forward, jumping, lifting the limbs, lateral flexion, vocalization, stepping, urination and head tremors). The rating was 0 (0 behaviors shown), 1 (1–2 behaviors shown), 2 (3–4 behaviors shown) or 3 (≥5 behaviors shown) (the sheet with the sedation and excitation scales can be found in the Appendix A). Both parameters were monitored every 5 min, during the whole experiment. Continuous video recording was performed during the whole experimental session.

### 2.6. Statistical Analysis

A Kolmogorov–Smirnov test was used to test the data distribution. Descriptive statistics for TCRt, HR, RR, sedation and excitation scores are presented as median and interquartile ranges (25% and 75%). No measurements after 120 min of injection were considered for statistical analysis, since longer data collection was performed only in three animals. For the temporal profiling of drug effects, five time windows of 30 min each were defined: BL (30 min pre-injection), T1 (0–30 min post-injection (PI)), T2 (31–60 min PI), T3 (61–90 min PI) and T4 (91–120 min PI). For comparisons between groups, the Mann–Whitney U test was used; for comparison within groups, the Friedman test with the Tukey–Kramer multiple comparison test was employed. The statistical significance level was set at 5% (*p* < 0.05) for comparisons between groups (Mann–Whitney U) and the Friedman test, and at 1% (*p* < 0.01) for the Tukey–Kramer multiple comparison test, according to the Bonferroni correction. For TCRt, HR and RR, the percent change between each time window and baseline was also calculated. The following formula was applied: percent (%) change = [(value calf_X_ T_X_ − value calf_X_ BL)/(value calf_X_ BL)] × 100. Afterwards, the median of the percent (%) changes was calculated for each time window (T1, T2, T3 and T4). A percent (%) change of >±20% was considered clinically relevant. The number of calves showing clinically relevant changes in TCRt, HR and RR was calculated, and reported as relative frequencies and percentages. Furthermore, the onset and offset of antinociceptive action were investigated. Onset of treatment action was defined as the time elapsed between drug administration and the timepoint at which the TCRt became 20% higher than the baseline and remained above this value for at least 5 min; offset was defined as the time point at which the TCRt came back to less than 20% over the baseline and remained below this value for at least 5 min. Onset of sedation and excitation was defined as the time between injection and the timepoint at which a score of > 1 was recorded and remained stable for at least 5 min. Maximum sedation and excitation was considered the highest cumulative score (sum of the scores for the 8 calves) at a given time point. 

Statistical analysis was performed using the software NCSS (NCSS Version 21.0.3, NCSS, LLC, Kaysville, UT, USA).

## 3. Results

All calves tolerated the experiment well and completed the study.

### 3.1. TCRt Measurement

At the baseline, no statistically significant difference between groups was present. A statistically significant difference between groups emerged at T1 (*p* < 0.01), T2 (*p* = 0.01) and T4 (*p* = 0.049). Similarly, the percent change from the baseline differed between groups for each post-treatment time window (*p* < 0.01). Within groups, a statistically significant increase in TCRt was observed between the baseline and T1, T2, T3 and T4 in the DB group (*p* < 0.01 for all differences), and between the baseline and T2 and T3 in the SB group (*p* < 0.01 for all differences). 

All the calves in the DB group had a clinically relevant increase in TCRt, that started at T1 and was maintained for the whole experiment. In the SB group, 50% of the calves had a clinically relevant increase in TCRt at T1, of 88% at T2 and T3, and of 75% at T4. 

The onset of treatment action was significantly different between groups (*p* = 0.04), being 0 (0–1.75) minutes in the DB group and 7 (0.75–31.25) minutes in the SB group. Similarly, the offset of treatment action differed for the two groups (*p* = 0.03), occurring at 151.5 (132.5–154.75) minutes in the DB group and at 90 (42–153) minutes in the SB group.

The complete TCRt results are reported in Table 1 and Figure 2. The TCRt representation over time (120 min) is presented in the Appendix A). 

### 3.2. Heart Rate and Respiratory Rate

For HR, no statistically significant difference was found between the DB and SB groups at the baseline. A difference emerged at T1 (*p* < 0.01), T2 (*p* < 0.01) and T3 (*p* = 0.04). The percent change from the baseline was also statistically different between groups at all the time windows (*p* < 0.01). Within groups, a decrease in HR (*p* < 0.01) was observed in the DB group between the baseline and T1, T2 and T3, as well as between T1 and T2, and T3 and T4. On the contrary, in the SB group, HR tended to increase, but the change was not statistically significant (*p* = 0.02). 

In the DB group, 88% of the calves had a clinically relevant decrease in HR at T1, while only 13% of the calves had this at T2, T3 and T4. In the SB group, 25% of the calves had a clinically relevant increase in HR at T1, 50% had this at T2, 38% had this at T3 and 25% had this at T4. 

For RR, no statistically significant differences were seen between the groups at the baseline. A statistically significant difference emerged between groups only at T2 (*p* = 0.04). For each time window, the percent change from the baseline did not differ between groups and no differences were found within groups. The complete HR and RR results are reported in Table 2.

### 3.3. Sedation and Excitation

For the sedation score, no statistically significant differences were found between groups at the baseline. A statistically significant difference between groups was found at T1 (*p* < 0.01), T2 (*p* < 0.01), T3 (*p* < 0.01) and T4 (*p* = 0.03). Within the DB group, a statistically significant difference (*p* < 0.01) was observed between the baseline and T1, T2 and T3, as well as between T4 and T1, T2 and T3. On the contrary, within the SB group, no difference (*p* = 0.95) in the sedation score was found over time. 

In the DB group, the onset of sedation was <1 min for four calves and 3 min for the other four. The maximal sedation score was registered at 5 min after injection. Three calves of the SB group sporadically showed a sedation score of 1 during the experiment (Figure 3A).

For the excitation score, no statistically significant differences were seen between the groups at the baseline. A statistically significant difference between the groups was found at T1 (*p* < 0.01), T2 (*p* < 0.01) and T4 (*p* < 0.01). Within the groups, statistically significant differences were found for the SB group (*p* < 0.01) between the baseline and T1 and T2, but not for the DB group (*p* = 0.95). 

In the SB group, 15 min after treatment administration, all calves showed some degree of excitation (≥1) and maximal excitation scores were registered 20 min after treatment administration. Moreover, some sporadic peaks in excitation were seen (i.e., time point 95 and 110) (Figure 3D). The sedation and excitation scores are reported in Table 3 and Figure 3A–D.

## 4. Discussion

The results show that the administration of butorphanol in healthy calves has a significative antinociceptive effect, which is substantially enhanced when the opioid is combined with detomidine. In the calves included in the present study, both butorphanol alone at 0.1 mg kg^−1^ or combined with detomidine at 0.02 mg kg^−1^ induced a statistically significant increase in the TCRt compared to that at the baseline. Heart rate significantly decreased compared to that at the baseline when butorphanol was combined with detomidine, but not when administered alone. Despite individual differences, calves receiving butorphanol alone were markedly less sedated and more excited than those receiving butorphanol and detomidine. 

### 4.1. Analgesia

Several studies on cattle have investigated the analgesic effects of alpha-2 agonists when administered alone or as part of drug combinations with or without butorphanol, and via different routes [6,9,27,28]. On the contrary, data on the analgesic efficacy of butorphanol alone are missing. Although some analgesic effect has been already reported, most of the studies agree that those protocols do not provide sufficient analgesia for the procedures performed (castration, disbudding, and claw treatment) [6,9,27,28]. In the present study, a mild antinociceptive effect arising from the administration of butorphanol was found, which is likely not sufficient to prevent pain during invasive procedures. On the contrary, the coadministration of detomidine at the reported dose further increased the nociceptive threshold, reaching levels that would suggest appropriate analgesia for minor interventions. Indeed, a similar extent of antinociceptive activity has been shown for clinically proven dosing regimens by previous studies on horses [23] and dogs [29].

### 4.2. Techniques to Assess Nociception

Regarding calves, there are several methods described in the literature for the assessment of pain and nociception. Plasma cortisol levels [5,6,7], responses to pinprick stimulation [13,15], electrodermal activity [7,30] and physiological parameters such as HR or food intake [5] are the most commonly described ones, but none can be considered specific [6,7]. In order to objectify the assessment of nociception, electrically evoked nociceptive reflexes have been proposed in veterinary research [18,20,22,23,31,32,33]. There are two main reflexes, the trigeminocervical reflex (TCR) and the nociceptive withdrawal reflex (NWR), the latter being the most frequently reported in veterinary literature [18,20,22,23,31,32,33]. The stimulus is easily controlled, reproducible and noninvasive and the output is a well-defined neurophysiological outcome [34]. The drawback of this methodology is that the applied current bypasses the peripheral nociceptors, thus not being directly comparable to any naturally occurring noxious stimulus. Despite this limitation, the continuous tracking of TCR or NWR thresholds as real-time indices of pharmacologically induced antinociceptive activity can provide interesting insights into the profiling of drug action. As limb and body movements occurring in the case of excitation could interfere with the EMG recording, TCRt was preferred as a main outcome for the present study. 

The TCR Is the response of neck muscles to the stimulation of the cutaneous area innervated by the trigeminal nerve. Its afferent pathway is formed by the sensory branches of the trigeminal nerve and the efferent arch is provided by cervical nerve motor fibers [21]. In humans, it has been shown to have three components, two early ones elicited by non-noxious stimuli (C1 and C2), and a third late one (C3) caused by noxious stimuli such as electrical stimulations [35]. In both humans [35] and horses [21], the onset of C3 has been found to occur approximately 50 ms and 70 ms after the stimulus, respectively. Based on this, the time window of interest for quantifying TCRt in the present study was set between 50 and 120, and it gave consistent results. 

### 4.3. Heart Rate and Respiratory Rate

According to Lin et al., following a 0.01 mg kg^−1^ detomidine IV and 0.05 mg kg^−1^ butorphanol, a statistically significant decrease in heart rate was observed 10 min after treatment administration and lasted for 50 min [4]. In our study, using double those doses, a clinically relevant decrease in heart rate was seen right after treatment administration, and lasted for 30 min. On the contrary, no statistically significant changes were found in the SB group. 

A significative decline in RR was described in studies using alpha-2 agonists as part of their protocols [3,4,26], while a significative increase was seen when butorphanol was injected in the subarachnoidal space [13]. Even though the different doses and routes impede a direct comparison, our results show the opposite, since no statistically significant changes in RR were seen in both groups.

### 4.4. Sedation and Excitation

Two previous studies investigated the sedative effects of the detomidine–butorphanol combination in calves. [3,4]. According to Lin et al., a 0.01 mg kg^−1^ detomidine and 0.05 mg kg^−1^ butorphanol IV induced sedation, which became apparent 10 min after injection and lasted for 43 min. In another recent study, a detomidine (0.03 mg kg^−1^) and butorphanol (0.1 mg kg^−1^) IV induced sedation in 42 s and this effect lasted for 58 min [3]. In the present study, maximal sedation was evident 5 min post-injection and was maintained at a clinically relevant level for approximately 90 min. No excitation was observed when calves received butorphanol combined with detomidine.

No previous studies have investigated sedative and/or excitatory effects of butorphanol alone in calves. In piglets receiving butorphanol at the dose of 0.2 mg kg^−1^ IM, statistically significant behavioral changes consistent with excitation were reported [14]. This is in concordance with our study where butorphanol administered alone caused a clinically relevant increase in the excitation score that peaked approximately 20 min after drug administration.

### 4.5. Limitations

No placebo group was used, but each patient acted as their own control. For that, a 30 min baseline was used. The presence of a placebo group might have contributed to differentiating drug-induced antinociception from the occurrence of natural transitory inhibitory states such as sleep or distraction, that could also alter nociceptive reflex thresholds and might be partially responsible for the results obtained. Furthermore, the inclusion of a saline–detomidine group would have allowed the determination of the efficacy of detomidine alone, thus highlighting the true contribution of butorphanol to the overall antinociceptive effect observed. Furthermore, the appearance of excitative behaviors could have also been influenced by the stress caused by prolonged restrain or hunger. 

## 5. Conclusions

The present findings can contribute to optimizing the application of these drugs in clinical practice. Butorphanol showed mild antinociceptive effects, which were enhanced when it was combined with detomidine. Calves receiving butorphanol alone showed marked excitation, which limits its applicability in clinical settings under the studied dosing regimen. The co-administration of detomidine was sufficient to eliminate the excitatory effect and induce consistent sedation in all the calves. 

## Figures and Tables

**Figure 1 animals-13-01943-f001:**
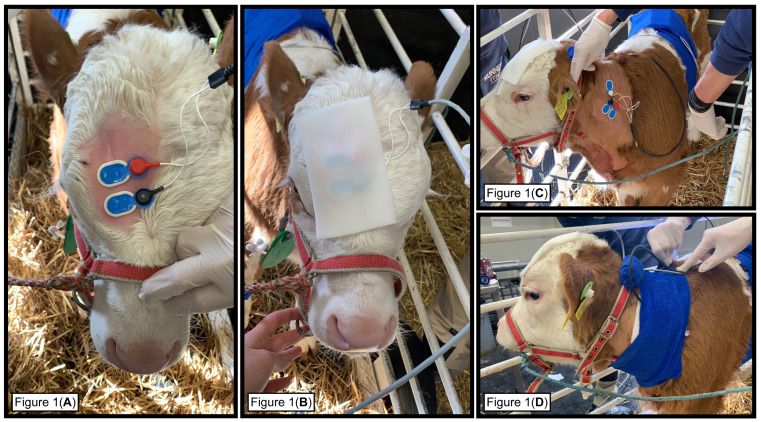
Stimulation electrodes, cathode (red) placed over the right supraorbital foramen and the anode (black) positioned approximately 1 cm rostrally to it (**A**). Latex-free self-adhesive foam dressing placed over stimulating electrodes and cables (**B**). Electromyography electrodes placed on the contralateral cleidoccipitalis muscle (red and black), and the ground electrode (white) positioned approximately 50 mm dorsally to the EMG electrodes (**C**). Self-adhesive foam and cross-elastic cohesive bandage placed around the neck (**D**).

**Figure 2 animals-13-01943-f002:**
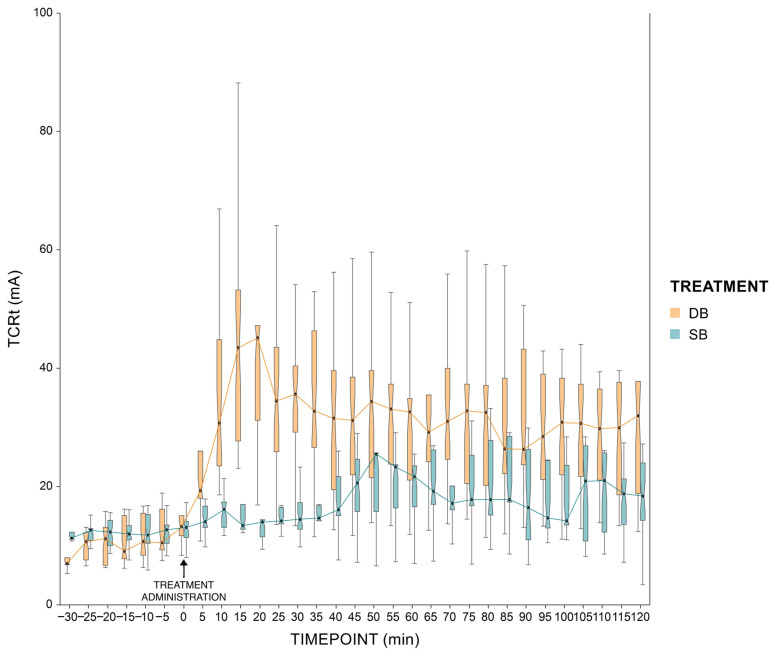
Box plot representing the median and interquartile ranges (25% and 75%) for the trigeminocervical reflex threshold over a 5 min interval, from the baseline up to 120 min. DB: detomidine–butorphanol (*n* = 8). SB: saline–butorphanol (*n* = 8). TCRt: trigeminocervical reflex threshold.

**Figure 3 animals-13-01943-f003:**
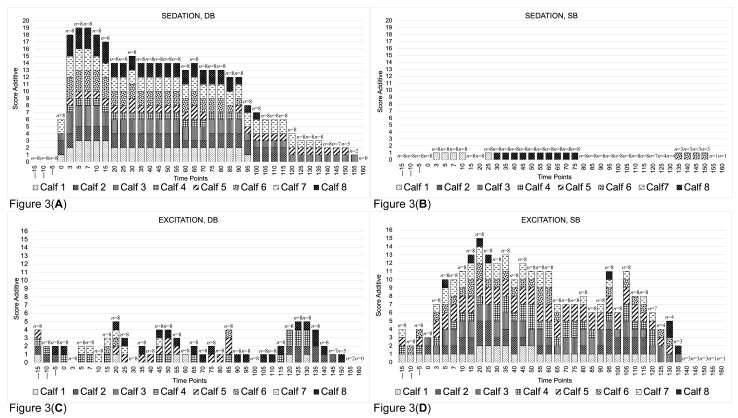
Bar graphs showing the sum of the sedation scores from each calf for DB (**A**). Bar graphs showing the sum of the sedation scores from each calf for SB (**B**). Bar graphs showing the sum of the excitation scores from each calf for DB (**C**). Bar graphs showing the sum of the excitation scores from each calf for SB (**D**). Data were collected and are presented in 5 min intervals.

**Table 1 animals-13-01943-t001:** Trigeminocervical Reflex threshold expressed as median and interquartile ranges for both groups, and for the five time windows. The percentage change with respect to the baseline, as well as the relative frequencies and percentages of calves with a clinically significative increase in the trigeminocervical reflex threshold are reported.

	Median (IQR) TCRt (mA)			Percentage (%) Change Respect to Baseline			Calves with Change of ≥20%	
	DB	SB	*p* Value ^§^	DB	SB	*p* Value ^§^	DB	SB
BL	10.5 (7.2–14.6)	12.4 (11.4–14.7)	0.23					
T1	34 (25.3–41.3)	14.2 (11.7–15.1)	<0.01	239%	22%	<0.01	8 (100%)	4 (50%)
T2	32.6 (12.6–37.5)	23.5 (16.1–23.9)	0.01	187%	48%	<0.01	8 (100%)	7 (88%)
T3	29.5 (22.8–36.9)	17.8 (15.3–28.2)	0.064	202%	47%	<0.01	8 (100%)	7 (88%)
T4	30.9 (21.3–37.6)	18.3 (12.8–26.5)	0.049	186%	51%	0.01	8 (100%)	6 (75%)
*p* value *	<0.01 **	<0.01 ***						

BL: baseline. DB: detomidine–butorphanol group (*n* = 8). SB: saline–butorphanol group (*n* = 8). TCRt: trigeminocervical reflex threshold. IQR: Interquartile ranges. ^§^ Mann–Whitney U test; * Friedman test; ** Tukey–Kramer multiple comparison test: *p* < 0.01 between BL and T1, T2, T3, and T4; *** Tukey–Kramer multiple comparison test: *p* < 0.01 between BL and T2 and T3.

**Table 2 animals-13-01943-t002:** Heart and respiratory rates expressed as median and interquartile ranges, for both groups and for the five time windows. The percentage change with respect to the baseline, as well as the relative frequencies and percentages of calves with a clinically significative increase in heart and respiratory rates are reported.

	Median (IQR) HR and RR (Beats and Breaths per Minute)			Percentage (%) Change Respect to Baseline			Calves with Change of ≥20%	
	DB	SB	*p* Value ^§^	DB	SB	*p* Value ^§^	DB	SB
HR BL	119 (114–136)	124 (108–143)	0.5					
HR T1	96 (92–100)	152 (112–154)	<0.01	−23%	6%	<0.01	7 (88%)	2 (25%)
HR T2	102 (101–112)	140 (118–156)	<0.01	−14%	17%	<0.01	1 (13%)	4 (50%)
HR T3	112 (103–117)	142 (108–162)	0.04	−10%	13%	<0.01	1 (13%)	3 (38%)
HR T4	111 (104–122)	142 (102–147)	0.07	−9%	6%	0.01	1 (13%)	2 (25%)
*p* value *	<0.01 **	0.02 ***						
RR BL	36 (34–36)	37 (30–44)	0.46					
RR T1	29 (24–44)	48 (32–54)	0.082	−19%	16%	0.1	6 (75%)	4 (50%)
RR T2	38 (32–40)	42 (42–44)	0.04	1%	10%	0.2	4 (50%)	3 (38%)
RR T3	40 (32–54)	45 (42–48)	0.34	2%	14%	0.6	4 (50%)	4 (50%)
RR T4	44 (32–60)	40 (36–44)	0.46	−1%	10%	0.5	4 (50%)	4 (50%)
*p* Value *	0.76	0.24						

BL: baseline. DB: detomidine–butorphanol group (*n* = 8). SB: saline–butorphanol group (*n* = 8). HR: heart rate. RR: respiratory rate. TCRt: trigeminocervical reflex threshold. IQR: interquartile ranges. ^§^ Mann–Whitney U test; * Friedman test; ** Tukey–Kramer multiple comparison test: *p* < 0.01 between BL and T1, T2, and T3; between T1 and T2, T3, and T4; *** Tukey–Kramer multiple comparison test: *p* < 0.01 between BL and T2 and T3.

**Table 3 animals-13-01943-t003:** Sedation and excitation scores expressed as median and interquartile ranges for both groups and for the five time windows.

	Median (IQR) Sedation and Excitation Scores		
	DB	SB	*p* Value ^§^
Sedation BL	0 (0–0)	0 (0–0)	1
Sedation T1	2 (1–3)	0 (0–0)	<0.01
Sedation T2	2 (1–2)	0 (0–0)	<0.01
Sedation T3	2 (1–2)	0 (0–0)	<0.01
Sedation T4	1 (0–2)	0 (0–0)	0.03
*p* value *	<0.01 **	0.95	
Excitation BL	1 (0–1)	1 (0–1)	0.63
Excitation T1	0 (0–0)	2 (1–2)	<0.01
Excitation T2	0 (0–1)	2 (1–2)	<0.01
Excitation T3	0 (0–1)	1 (0–2)	0.1
Excitation T4	0 (0–0)	1 (1–1)	<0.01
*p* value *	0.95	<0.01 ***	

BL: baseline. DB: detomidine–butorphanol group (*n* = 8). SB: saline-butorphanol group (*n* = 8). IQR: interquartile ranges. ^§^ Mann–Whitney U test; * Friedman test; ** Tukey–Kramer multiple comparison test: *p* < 0.01 between BL and T1, T2, and T3, and between T4 and T1, T2, and T3; *** Tukey–Kramer multiple comparison test: *p* < 0.01 between BL and T1 and T2.

## Data Availability

Data supporting the reported results can be provided in case of interest upon request.

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
