# Peer review of "Antinociceptive, Sedative and Excitatory Effects of Intravenous Butorphanol Administered Alone or in Combination with Detomidine in Calves: A Prospective, Randomized, Blinded Cross-Over Study"

_animals, 2023, doi:10.3390/ani13121943_

Round 1
Reviewer 1 Report
Dear authors,
Many thanks for this submission. Very few comments to make, this is an incredibly well presented paper in my opinion on a subject that has previously me given significant headaches trying to understand it.
L87: I suppose you need to specify somehow that you are reporting these demographics as a median (min-max) - not a mean.
L141: please define, explain or clarify the notion of peak Z score as well as your calculation for readers unfamiliar with the equipment and the technique used. it also seems that this parameter is not mentioned anywhere else in the paper?
Figure 2: I cannot see where DB and SB appear in the graph. Based on the text I can guess which curve corresponds to which group, but I think there should be a better way of seeing this.
Very good overall.
Author Response
Dear reviewer,
Thank you very much for your comments and suggestions.
Please find below the answers to your comments.
L87: I suppose you need to specify somehow that you are reporting these demographics as a median (min-max) - not a mean.
Changed accordingly in the line 96.
L141: please define, explain or clarify the notion of peak Z score as well as your calculation for readers unfamiliar with the equipment and the technique used. it also seems that this parameter is not mentioned anywhere else in the paper?
Thank you very much for your comment. A clarification has been now included in the text between lines 278-282
Indeed, "Peak Z Score" is one of the several methods available to the user of the Dolosys to define the reflex "size" that the machine will interpreted as threshold. As it has to be set at the beginning of the experiment and it remains the same for all the recordings, we report it to allow reproducibility of the study.
Figure 2: I cannot see where DB and SB appear in the graph. Based on the text I can guess which curve corresponds to which group, but I think there should be a better way of seeing this.
Thank you very much for this comment. This should be an error of formatting from my part, the graph was moved too much to the right and the part of the legend got cut out. I hope now it is visible.
Reviewer 2 Report
Dear editor and authors,
Thank you for choosing me to be part of this publication process. It is an interesting study, trying to add information about analgesia in ruminants, sometime considered a “hidden subject”.
It is a well-designed study, describing a good methodology to access analgesia (antinociception) and behaviour effects of butorphanol or butorphanol-detomidine in calves. Please, find below some minor concerns about it that I hope, could contribute to the final version of this paper.
Introduction
- lines 51-52: According to the text, this sentence should be about “detomidine effects”. So, refs #8 and #11 are not about it.
- lines 54-55: Please, pay attention on the refs. I think ref #15 has no relation with the goal of this study. Different from the ref #3, which is in line with. I suggest to remove refs that have no relation or strict focus with this study (as refs #8, #11, #13, for instance).
- lines 58-59: Again, focus on the study. Think about to put here all of the studies that have shown excitation in many species... not fair.
- lines 60-70: Why do you not consider mechanical and electrical stimuli as “nociceptive stimuli”? I’m not sure about it…
Materials and Methods
- line 86: Which breed are they?
- lines 174-175: I think you can remove “For each time… and excitation scores”; a bit redundant information.
Results
- lines 233-23: Why did you choose 0.01 to be considered significant? It is fair enough to consider 0.05 here, founding statistical difference, which is easy "to see"...
- line 243: Remove this line.
- line 273: Remove this line.
Discussion
- lines 284-286: There are good results here, but a good question is "would detomidine alone promote the same effect as the DB group? It should be very important to perform a detomidine-saline (DS) group. Who knows the DS effects are the same or differect to the DB? I mean, It was evident detomidine-butorphanol would promote very good antinociceptive and sedative effects. But what about detomidine alone? Maybe butorphanol is necessary... or not... tricky
- line 296: I have no access to the ref #28, unfortunately, and I guess what its importance to the context… I think it could be removed from the paper.
- lines 297-298: “has been already reported” instead of “was reported in the about cited literature”.
- lines 303-304: It's quite important to set which mA is compared to minor interventions. I think 50mA mimics skin incision, but I’m not sure about it… Besides, what kind of interventions do you consider to be "minor"?
- 4.3 Heart rate and Respiratory rate: Please, remove results description here. It is too redundant.
- lines 345-351: This part is just descriptive. You should compare your results to the previous ones.
- 4.5. Limitations: You should consider the lack of detomidine-saline group, as commented before.
Conclusions
- Suggestion: Start the conclusion with “The present findings can contribute 374 to optimize the application of these drugs in the clinical practice.”, and then, “Butorphanol….”. Remove “strongly”, because it is too “strong” here…
- Remove “This study provides quantitative evidence of the efficacy of butorphanol administered alone or combined with detomidine in calves.”
Author Response
Dear reviewer,
Thank you very much for your comments and suggestions.
Please find below the answers to your comments.
Introduction
- lines 51-52: According to the text, this sentence should be about “detomidine effects”. So, refs #8 and #11 are not about it.
References eliminated and changed accordingly in the main text in line 60.
- lines 54-55: Please, pay attention on the refs. I think ref #15 has no relation with the goal of this study. Different from the ref #3, which is in line with. I suggest to remove refs that have no relation or strict focus with this study (as refs #8, #11, #13, for instance).
Thank you for the suggestion, some references have been modified accordingly. Nevertheless we believe that some of the cited references (8 and 13) still are pertinent to our study and were therefore kept.
- lines 58-59: Again, focus on the study. Think about to put here all of the studies that have shown excitation in many species... not fair.
Thank you very much for your comment. Despite understanding your point, we included the reference in pigs to justify our hypothesis (butorphanol induces excitation). Indeed, no information in calves is available.
- lines 60-70: Why do you not consider mechanical and electrical stimuli as “nociceptive stimuli”? I’m not sure about it…
Thank you very much for this comment. I believe there was a misunderstanding/wrong sentence formulation. The focus of this paragraph is the difference between the assessment of a behavioural response versus the tracking of the nociceptive reflexes thresholds. The text has been modified accordingly between lines 69-81
Materials and Methods
- line 86: Which breed are they?
Changed accordingly in the text in lines 97-98.
- lines 174-175: I think you can remove “For each time… and excitation scores”; a bit redundant information.
Changed accordingly in the text in line 366.
Results
- lines 233-23: Why did you choose 0.01 to be considered significant? It is fair enough to consider 0.05 here, founding statistical difference, which is easy "to see"...
Thank you very much for this comment. We set up the significance level at 1% for the Tukey Kramer multiple comparison test, as Bonferroni correction was used. This approach was suggested by the statistician that reviewed our work.
- line 243: Remove this line.
Thank you for your comment. Unfortunately, we cannot remove it because the journal guidelines states that in order to report values in a table, it should be also cited in the text.
- line 273: Remove this line.
See previous comment.
Discussion
- lines 284-286: There are good results here, but a good question is "would detomidine alone promote the same effect as the DB group? It should be very important to perform a detomidine-saline (DS) group. Who knows the DS effects are the same or differect to the DB? I mean, It was evident detomidine-butorphanol would promote very good antinociceptive and sedative effects. But what about detomidine alone? Maybe butorphanol is necessary... or not... tricky
Thank you very much for this comment. We are very aware of this point, and it has been (as suggested later) included in the limitations of the study.
- line 296: I have no access to the ref #28, unfortunately, and I guess what its importance to the context… I think it could be removed from the paper.
Changed accordingly in the text and reference eliminated in line 527.
- lines 297-298: “has been already reported” instead of “was reported in the about cited literature”.
Changed accordingly in the text in line 515.
- lines 303-304: It's quite important to set which mA is compared to minor interventions. I think 50mA mimics skin incision, but I’m not sure about it… Besides, what kind of interventions do you consider to be "minor"?
Electrical stimulation is unfortunately quite different from a "true" painful stimulation, as it bypasses the cutaneous nociceptors and directly activates the peripheral nerves. The sensation it evokes is nevertheless painful if the stimulus is strong enough to evoke withdrawal (recorded by EMG). In humans values of around 10 mA are perceived as painful. A stimulation of 50 mA is strongly painful but of very short duration and with no after sensation. Based on our experience in other species, a 2-3 times increase in nociceptive reflex threshold compared to baseline (awake, non medicated animal) normally allows procedures like wound revisions, skin suture, casting etc. For more invasive procedures, general or loco-regional anaesthesia needs to be added..
- 4.3 Heart rate and Respiratory rate: Please, remove results description here. It is too redundant.
Changed accordingly in the text in line 551.
- lines 345-351: This part is just descriptive. You should compare your results to the previous ones.
Changed accordingly in the text in lines 551-731.
- 4.5. Limitations: You should consider the lack of detomidine-saline group, as commented before.
Changed accordingly in the text in lines 748-756.
Conclusions
- Suggestion: Start the conclusion with “The present findings can contribute 374 to optimize the application of these drugs in the clinical practice.”, and then, “Butorphanol….”. Remove “strongly”, because it is too “strong” here…
Changed accordingly in the text in line 759.
- Remove “This study provides quantitative evidence of the efficacy of butorphanol administered alone or combined with detomidine in calves.”
Changed accordingly in the text.
Reviewer 3 Report
Reviewer comments on Manuscript number: animals-2356174
The present manuscript showed the anti-nociceptive, sedative and excitatory effect of a single dose of butorphanol with or without detomidine in calves.
Broad comments
The data showed is potentially useful, as there is a lack of current literature about the anti-nociceptive, sedative and excitatory effect of butorphanol as part of injectable sedation and general anesthesia protocols in calves; the manuscript in well written and just some details are needed to address in order to be published.
Weakness of the study: Lack of placebo group.
Specific comments
L2–5. Title. To this reviewer the term “behavioral” is too wide and the study design does not support its whole approach. “Sedative and excitatory” instead “behavioral” effect could be better described at this end.
L40. Keywords. Please replace analgesia with anti-nociception.
Introduction.
L55. Please complete the idea “…… that adding butorphanol to……”
Material and methods.
L90. Please add the ethical committee number approval for this study. And please add all the missing information considering CONSORT guidelines.
L126. Even when the main aim of the study was the anti-nociceptive effect, the authors may also calculate the sample size considering the expected differences (differences among means) in heart rate between groups.
L134. Please add the specific information about drugs used, trade mark, country of manufacturing, drug concentration.
L156. Please specify the ECG lead that was continuously recorded.
L158. Please specify if the thoracic excursions were counted per minute.
L162. Please specify the clinical signs that were included in each ratting (0, 1, 2 and 3).
L164. Idem above for excitation.
L165. Is there a link to watch the recorded videos?
Results.
L235. If the p value is correct (p=0.02), then the HR change was statically significant.
L237. Please specify what does “clinically relevant” mean.
Discussion.
L313. NWR abbreviations have not been described before.
Thanks
Author Response
Dear reviewer,
Thank you very much for your comments and suggestions.
Please find below the answers to your comments.
• L2–5. Title.To this reviewer the term “behavioral” is too wide and the study design does not support its whole approach. “Sedative and excitatory” instead “behavioral” effect could be better described at this end.
Changed accordingly in the text in line 2.
• L40.Keywords. Please replace analgesia with anti-nociception.
Changed accordingly in the text line 40.
Introduction.
• L55.Please complete the idea “…… that adding butorphanol to……”
Changed accordingly in the text in line 63.
Material and methods.
• L90. Please add the ethical committee number approval for this study. And please add all the missing information considering CONSORT guidelines.
Thank you for your comment. The number of ethical committee approval was included in line 100-102. Regarding the guidelines, since the study involved animals, the paper was written following the ARRIVE guidelines and those have been included in the new submission.
• L126. Even when the main aim of the study was the anti-nociceptive effect, the authors may also calculate the sample size considering the expected differences (differences among means) in heart rate between groups.
Thank you very much for this comment. As we mention in the section of materials and methods, facing the lack availability of species-specific data about the extent of antinociceptive effects of opioids in calves, a sample size equal to the one used in Studer et al. was applied and no calculation was done. As HR was not a main outcome variable, we did not consider appropriate to use it for the sample size calculation.
• L134.Please add the specific information about drugs used, trade mark, country of manufacturing, drug concentration.
Changed accordingly in lines 263-265.
• L156. Please specify the ECG lead that was continuously recorded.
Changed accordingly in line 290.
• L158. Please specify if the thoracic excursions were counted per minute.
Changed accordingly in line 292.
• L162. Please specify the clinical signs that were included in each ratting (0, 1, 2 and 3).
Changed accordingly in line 297-299
• L164. Idem above for excitation.
Changed accordingly in line 353-356
• L165. Is there a link to watch the recorded videos?
There are not, but the material can be available by request to the authors.
Results.
• L235. If the p value is correct (p=0.02), then the HR change was statically significant.
Thank you very much for your comment. When doing the comparison within group, we used the Tukey-Kramer multiple comparison test, corrected with the Bonferroni method, and therefore we used a level of significance of 1%.
• L237. Please specify what does “clinically relevant” mean.
Thank you very much for your comment. The concept of clinically relevant is explained on line 375, in the section of Statistical Analysis.
Discussion.
• L313. NWR abbreviations have not been described before.
Thank you very much for your comment. This abbreviation was cited in line 258 in the section of Sample Size calculation. Nevertheless, we included now the complete wording in the discussion.
